# Flow environment and matrix structure interact to determine spatial competition in *Pseudomonas aeruginosa* biofilms

Carey D Nadell[1†], Deirdre Ricaurte[2†], Jing Yan[2], Knut Drescher[1], Bonnie L Bassler[2,3]*

[1]Max Planck Institute for Terrestrial Microbiology, Marburg, Germany; [2]Department of Molecular Biology, Princeton University, Princeton, United States; [3]Howard Hughes Medical Institute, Chevy Chase, United States

**Abstract** Bacteria often live in biofilms, which are microbial communities surrounded by a secreted extracellular matrix. Here, we demonstrate that hydrodynamic flow and matrix organization interact to shape competitive dynamics in *Pseudomonas aeruginosa* biofilms. Irrespective of initial frequency, in competition with matrix mutants, wild-type cells always increase in relative abundance in planar microfluidic devices under simple flow regimes. By contrast, in microenvironments with complex, irregular flow profiles – which are common in natural environments – wild-type matrix-producing and isogenic non-producing strains can coexist. This result stems from local obstruction of flow by wild-type matrix producers, which generates regions of near-zero shear that allow matrix mutants to locally accumulate. Our findings connect the evolutionary stability of matrix production with the hydrodynamics and spatial structure of the surrounding environment, providing a potential explanation for the variation in biofilm matrix secretion observed among bacteria in natural environments.

*For correspondence: bbassler@ princeton.edu

†These authors contributed equally to this work

**Competing interests:** The authors declare that no competing interests exist.

## Introduction

In nature, bacteria predominantly exist in biofilms, which are surface-attached or free-floating communities of cells held together by a secreted matrix (*Hall-Stoodley et al., 2004*; *Nadell et al., 2009*; *Flemming et al., 2016*). The extracellular matrix defines biofilm structure, promotes cell-cell and cell-surface adhesion, and confers resistance to chemical and physical insults (*Flemming and Wingender, 2010*; *Hobley et al., 2015*; *Teschler et al., 2015*; *Tseng et al., 2013*; *Doroshenko et al., 2014*; *Landry et al., 2006*). The matrix also plays a role in the social evolution and population dynamics of biofilm-dwelling bacteria (*Nadell et al., 2009*; *Steenackers et al., 2016*; *Nadell et al., 2016*; *Ghoul and Mitri, 2016*; *Mitri et al., 2016*, *2013*, *2011*). In some species, such as the soil bacterium *Bacillus subtilis*, matrix materials are readily shared among cells, leading to public goods dilemmas in which non-producing strains can outcompete producing strains (*van Gestel et al., 2015*, *2014*; *Kovács, 2014*). In other species, including the pathogens *Vibrio cholerae*, *Pseudomonas fluorescens*, and *P. aeruginosa*, matrix-secreting cell lineages privatize most matrix components, allowing them to smother or laterally displace other cell lineages and, in so doing, outcompete non-producing cells (*Xavier and Foster, 2007*; *Nadell and Bassler, 2011*; *Nadell et al., 2015*; *Kim et al., 2014c*; *Schluter et al., 2015*; *Irie et al., 2016*; *Drescher et al., 2016*; *Yan et al., 2016*; *Madsen et al., 2015*; *Oliveira et al., 2015*). Interestingly, not all wild and clinical isolates of these species produce a biofilm matrix, despite the clear ecological and competitive benefits of possessing a matrix (*Mann and Wozniak, 2012*; *Yawata et al., 2014*; *Chowdhury et al., 2016*). Theory and experiments investigating bacterial colonies on agar show that constrained movement can promote

**eLife digest** Bacteria often live together – attached to surfaces like river rocks, water pipes, the lining of the gut and catheters – in communities called biofilms. These groups of bacteria are small-scale ecosystems in which cells cooperate and compete with one another to obtain resources, such as food and space to grow. Within a biofilm, a sticky glue-like substance called the matrix binds the cells to each other and to the surface. Cells that make the matrix typically have an advantage over those that do not because they can better resist the shearing forces experienced when liquid flows over the surface. The matrix also helps cells to capture nutrients from the passing liquid. Nevertheless, not all strains of bacteria make matrix, despite its advantages.

Because of where they can grow, biofilms are fundamentally important in the environment, in industry and in infections. Resolving why some bacteria make matrix while others do not could therefore allow scientists and engineers to re-design the surfaces involved in these settings to discourage harmful biofilms or to encourage beneficial ones.

Nadell, Ricaurte et al. have now used a bacterium called *Pseudomonas aeruginosa* to explore how the properties of the surface and the flowing liquid affect matrix production among cells in biofilms. *P. aeruginosa* typically lives in soil and can cause infections in people, especially in hospital patients and people who have weakened immune systems. Nadell, Ricaurte et al. studied normal *P. aeruginosa* bacteria and a mutant strain that is unable to make matrix. The strains were labeled with fluorescent markers and put into special chambers that simulated different environments. The proportion of each strain was measured after three days of biofilm growth. When biofilms were grown under flowing liquid in simple environments with flat surfaces, matrix producers always outcompeted non-producers. However, the two strains coexisted in more complex and porous environments, like those found in soil.

Nadell, Ricaurte et al. went on to show that the strains could co-exist because the matrix producers made biofilms that created areas within the environment where the liquid flows very slowly or not at all. In these regions, non-producing cells could compete successfully because resistance to shearing forces is less important when flow is weak or absent, and so the non-producing cells were not washed away. The results begin to explain why matrix production among cells in environmental settings is diverse and highlight that the environment is important in the evolution of bacterial biofilms.

coexistence of different strains and species (*Levin, 1974*; *Levin and Paine, 1974*; *Durrett and Levin, 1994*, *1998*; *Kerr et al., 2002*; *Kim et al., 2008*; *Poltak and Cooper, 2011*). Fitness trade-offs between the benefits of being adhered to surfaces and the ability to disperse to new locations can cause variability in matrix production (*Nadell and Bassler, 2011*; *Yawata et al., 2014*; *Levin, 1974*; *Cohen and Levin, 1991*), but it is not well understood how selective forces within the biofilm environment itself might drive the coexistence of strains that make matrix with strains that do not. Here, we explore how selection for matrix production occurs within biofilms on different surface geometries and under different flow regimes, including those that are relevant inside host organisms and in abiotic environments such as soil.

The local hydrodynamics associated with natural environments can have dramatic effects on biofilm matrix organization. This phenomenon has been particularly well established for *P. aeruginosa*, a common soil bacterium (*Green et al., 1974*) and opportunistic pathogen that thrives in open wounds (*Fazli et al., 2009*; *Burmølle et al., 2010*), on sub-epithelial medical devices (*Guaglianone et al., 2010*), and in the lungs of cystic fibrosis patients (*Harmsen et al., 2010*; *Ciofu et al., 2013*; *Folkesson et al., 2012*; *Stacy et al., 2015*; *McNally et al., 2014*). Under steady laminar flow in simple microfluidic channels, *P. aeruginosa* forms biofilms with intermittent mushroom-shaped tower structures (*Harmsen et al., 2010*; *Friedman and Kolter, 2004*; *Miller et al., 2012*; *Parsek and Tolker-Nielsen, 2008*). Under irregular flow regimes in more complex environments, however, *P. aeruginosa* also produces sieve-like biofilm streamers that protrude into the liquid phase above the substratum (*Persat et al., 2015*; *Kim et al., 2014a*, *2014b*; *Rusconi et al., 2010*). These streamers – whose structure depends on the secreted matrix – are proficient at

catching cells, nutrients, and debris that pass by, leading to clogging and termination of local flow (*Drescher et al., 2013*).

The spatial and temporal characteristics of flow thus combine to alter matrix morphology, which, in turn, feeds back to alter local hydrodynamics and nutrient advection and diffusion (*Nadell et al., 2016*; *Hellweger et al., 2016*; *Stewart and Franklin, 2008*). By modifying community structure and solute transport in and around biofilms (*Stewart, 2012*), this feedback could have a significant influence on the evolutionary dynamics of matrix secretion in natural environments (*Coyte et al., 2016*). Here, we study within-biofilm competition as a function of flow regime using strains of *P. aeruginosa* PA14 that differ only in their production of Pel, a viscoelastic matrix polysaccharide that serves as the primary structural element for biofilm and streamer formation (*Friedman and Kolter, 2004*; *Drescher et al., 2013*; *Chew et al., 2014*; *Jennings et al., 2015*). Using a combination of fluid flow visualization and population dynamics analyses, we reveal a novel interaction between hydrodynamic conditions, biofilm architecture, and competition within bacterial communities.

## Results/discussion

We performed competition experiments with wild type PA14 and an otherwise isogenic strain deleted for *pelA*, which is required for synthesis of Pel (*Franklin et al., 2011*). We focused on Pel because it is the key structural polysaccharide in PA14 biofilms, and it is necessary for streamer formation under complex flow regimes. Deletion of *pelA* significantly impairs biofilm formation in PA14, which does not naturally produce Psl, an additional matrix polysaccharide secreted by other *P. aeruginosa* isolates (*Colvin et al., 2012*, *2011*). Wild-type cells produced GFP, and Δ*pelA* mutants produced mCherry. Experiments in shaken liquid culture using genetically identical wild-type cells producing GFP or mCherry confirmed that fluorescent protein expression constructs had no measurable effect on growth rate (*Figure 1—figure supplement 1*). Our first goal was to compare the population dynamics of the wild-type and Δ*pelA* strains in typical planar microfluidic devices, which have simple parabolic flow regimes, and in porous environments containing turns and corners, which have irregular flow profiles and better reflect the packed soil environments that *P. aeruginosa* often occupies (*Green et al., 1974*; *Das and Mukherjee, 2007*; *Stover et al., 2000*). To approximate the latter environment, we used microfluidic chambers containing column obstacles. The size and spacing distributions of the column obstacles were specifically designed to simulate soil or sand (see Materials and methods). Analogous methods have been used previously to study bacterial growth (*Vos et al., 2013*) and the behavior of *Caenorhabditis elegans* (*Lockery et al., 2008*) in realistic environments while maintaining accessibility to microscopy. In our setup, flow was maintained through these chambers at rates comparable to those experienced by *P. aeruginosa* in a soil environment (*Heath, 1983*).

### The flow regime alters selection for matrix production in biofilms

Several approaches are available to study how competitive dynamics differ in particular flow environments. Most commonly, one would monitor biofilm co-cultures of wild-type and Δ*pelA* PA14 cells over time until their strain compositions reached steady state. Performing such time-series experiments was not possible here due to a combination of low-fluorescence output in the early phases of biofilm growth coupled with phototoxicity incurred by cells during epifluorescence imaging. To circumvent this issue, we measured the change in frequency of wild-type and Δ*pelA* cells over a fixed 72 hr time period as a function of their initial ratio in both planar chambers and soil-mimicking chambers containing column obstacles. Population composition was quantified in all cases using microscopy, as described in the Materials and methods section. From these measurements, we could infer the final stable states of Pel-producing and non-producing cells as a function of surface topography and flow conditions. This method is commonly used to evaluate the behavior of dynamical systems, and it has been employed in a variety of related experimental applications (*Nadell and Bassler, 2011*; *Nadell et al., 2015*; *Madsen et al., 2015*; *Chuang et al., 2009*; *Sanchez and Gore, 2013*; *Drescher et al., 2014*).

In planar chambers with simple parabolic flows, wild-type PA14 increased in relative abundance regardless of initial population composition, indicating uniform positive selection for Pel secretion (*Figure 1A,B*). This result is consistent with recent studies of *V. cholerae* and *Pseudomonas* spp. demonstrating that – in these species – core structural polysaccharides of the secreted matrix cannot

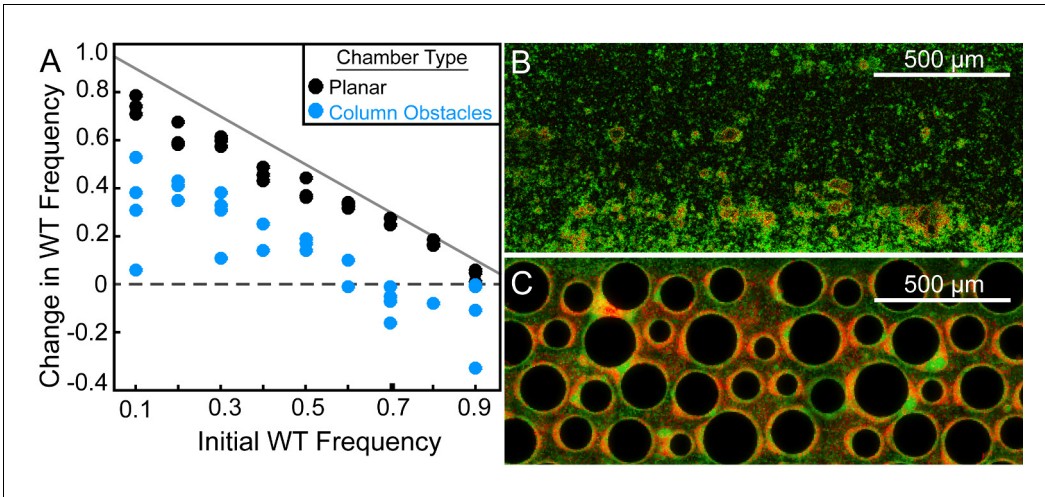

**Figure 1.** Wild-type *P. aeruginosa* PA14 outcompetes the *ΔpelA* mutant under simple flow conditions, but the two strains coexist under complex flow conditions. (A) Wild-type and *ΔpelA* strains were co-cultured at a range of initial frequencies in the simple planar (black data) or column-containing (blue data) microfluidic chambers. In both cases, fresh minimal M9 medium with 0.5% glucose was introduced at flow rates adjusted to equalize the volume of medium flowing through each chamber per unit time across all experiments. The diagonal gray line denotes the maximum possible increase in wild-type frequency for a given initial condition. Each data point is an independent biological replicate. (B) A maximum intensity projection (top-down view) of a confocal *z*-stack of wild type (green) and *ΔpelA* (red) biofilms in simple flow chambers. (C) An epifluorescence micrograph (top-down view) of wild type (green) and *ΔpelA* (red) biofilms after 72 hr growth in a flow chamber containing column obstacles to simulate a porous environment with irregular flows. Images in (B) and (C) were taken from chambers in which the wild type was inoculated at a frequency of 0.7.

The following source data and figure supplements are available for figure 1:

**Source data 1.** Change in WT frequency as a function of initial frequency in two flow conditions

**Figure supplement 1.** The maximum growth rates of *P. aeruginosa* PA14 wild type and *ΔpelA* cells in mixed liquid culture.

**Figure supplement 1—source data 1.** Maximum liquid culture growth rates of study strains.

be readily exploited by non-producing mutants (*Nadell and Bassler, 2011*; *Nadell et al., 2015*; *Kim et al., 2014c*; *Schluter et al., 2015*; *Irie et al., 2016*; *Yan et al., 2016*; *Madsen et al., 2015*). Confocal microscopy revealed that Pel-producers mostly excluded non-Pel-producing cells from biofilm clusters in planar chambers (*Figure 2A,B*), although some *ΔpelA* mutants resided on the periphery of wild-type biofilms. The liquid effluent from these chambers contained an over-representation of the *ΔpelA* mutant relative to wild type, consistent with the interpretation that *ΔpelA* strain was displaced from the substratum over time (*Figure 2C*). When wild-type and *ΔpelA* cells competed in microfluidic devices simulating porous microenvironments, by contrast, there was a pronounced shift to negative frequency-dependent selection for Pel production (*Figure 1A,C*). Wild-type PA14 was selectively favored at initial frequencies below ~0.6. Above this critical frequency, the *ΔpelA* mutant was favored. From this result, we can infer that in this porous environment, *ΔpelA* null mutants can grow and stably coexist with wild type Pel-producers.

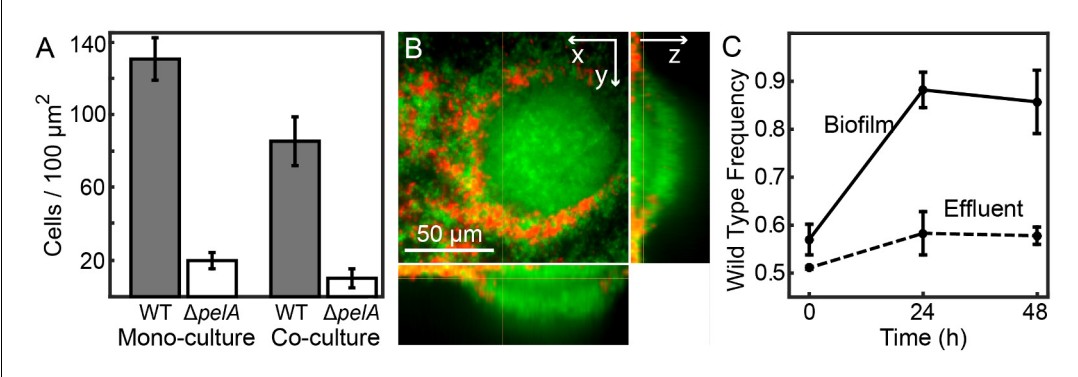

**Figure 2.** Matrix production confers a competitive advantage to wild-type *P. aeruginosa* PA14 in biofilms under simple flow conditions. (**A**) Absolute abundances of wild-type and *ΔpelA* strains in monoculture and co-culture in planar microfluidic flow-chambers (bars denote means ± S.D. for n = 3–6). The two strains were inoculated alone (left two bars) or together at a 1:1 ratio (right two bars). (**B**) Single optical plane 3 µm from the surface, and *z*-projections at right and bottom respectively, of the wild-type (green) and *ΔpelA* (red) strains grown in co-culture for 48 hr after inoculation at a 1:1 ratio. (**C**) Relative abundance of wild-type and *ΔpelA* strains in the liquid effluent of planar microfluidic devices (points denote means ± S.D. for n = 3). Wild-type and *ΔpelA* cells were combined at a 1:1 initial ratio and co-inoculated on the glass substratum of simple flow chambers. At 0, 24, and 48 hr, 5 µL of effluent was collected from chamber outlets. Wild-type frequency was calculated within biofilms and within the liquid effluent for each time point.

The following source data is available for figure 2:

**Source data 1.** Biofilm production of WT and Pel-deficient *P. aeruginosa* in mono-culture and co-culture; cell counts in chamber effluents.

## Wild-type *P. aeruginosa* obstructs porous environments, generating low-shear regions that Pel-deficient mutants can occupy

Previous work has shown that in environments with flow and with corners, wild-type *P. aeruginosa* produces Pel-dependent biofilm streamers that extrude from the surface into the passing liquid (*Kim et al., 2014a*; *Rusconi et al., 2010*; *Drescher et al., 2013*). In our experiments, streamers were produced by wild-type cells and could be readily detected via microscopy throughout column-containing chambers, but not planar chambers. Streamers are known to catch cells and debris that pass by (*Drescher et al., 2013*), and although *ΔpelA* cells could be found in the streamers in our experiments, they were not abundant. Cell capture by streamers therefore cannot account for the observed coexistence of the two strains (*Figure 3—figure supplement 1*). This result suggests that streamers do indeed catch debris, consistent with prior studies (*Kim et al., 2014b*; *Drescher et al., 2013*), but that in our system, *ΔpelA* cells are not present at high enough density in the passing liquid phase to accumulate substantial population sizes by this mechanism.

Our microscopy-based observation of chambers containing column obstacles suggested that wild-type biofilms gradually obstructed some of the regions located between columns over time. We hypothesized that partial clogging could render those portions of the chambers more suitable for growth of the *ΔpelA* strain, which was previously shown to be sensitive to removal by shear (*Colvin et al., 2012*, *2011*). This hypothesis predicts that *ΔpelA* cells should be found predominantly in regions of the chamber that have been clogged by wild type biofilms. To test this prediction, we repeated our co-culture competition experiment with wild-type and *ΔpelA* cells in chambers containing columns, and we measured the distribution of each strain as above. We next introduced fluorescent beads into the chambers by connecting new influent syringes to the inflow tubing. By tracking the beads with high frame-rate microscopy, we could distinguish areas in which flow was present from areas in which flow was absent or very low, and then we could superimpose this information onto the spatial distributions of wild-type and *ΔpelA* mutant cells (*Figure 3—figure supplement 2*).

Wild-type biofilms accumulated intermittently, often with clusters of *ΔpelA* cells in close proximity. Importantly, and in support of our prediction, *ΔpelA* biofilm clusters occurred significantly more often in regions in which flow was blocked by wild-type biofilms than in regions in which flow was not interrupted (*Figure 3A,B*). As shown previously (*Drescher et al., 2013*), *ΔpelA* cells did not clog chambers when grown in isolation, supporting the interpretation that *ΔpelA* accumulation relies on

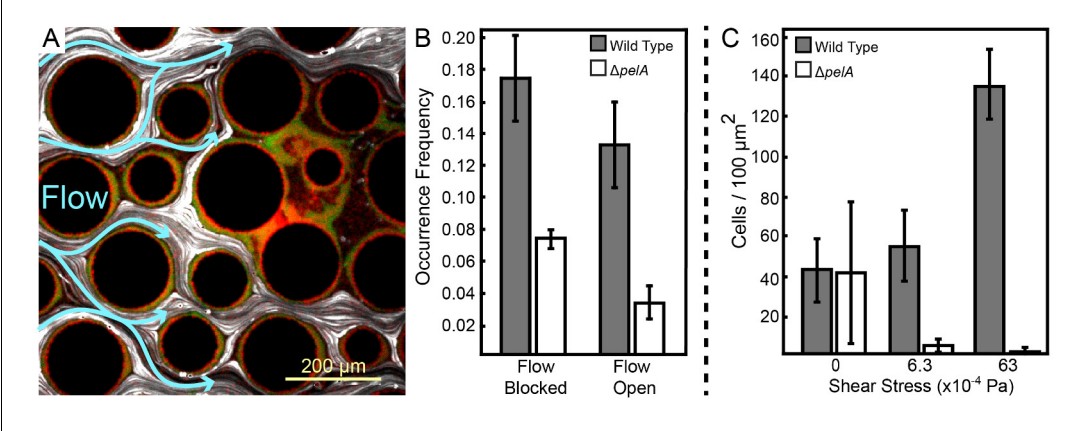

**Figure 3.** Pel-deficient mutants occupy locations protected from flow due to local clogging by wild-type *P. aeruginosa* biofilms. (A) Wild-type (green) and Δ*pelA* (red) *P. aeruginosa* strain mixtures were inoculated into complex flow chambers with irregularly-spaced column obstacles. Biofilms were imaged using confocal microscopy, after which fluorescent beads were flowed through the chamber. The presence or absence of flow was monitored through averaging successive exposures of bead tracks (white lines are bead tracks; blue arrows highlight flow trajectories). (B) Analysis of co-occurrence of flow and wild-type or Δ*pelA* cell growth at the end of 1:1 competition experiments in complex flow chambers with column obstacles, as illustrated by the micrograph in (A). The occurrence of wild-type (gray) and Δ*pelA* (white) cell clusters are shown as a function of whether local flow has been blocked or remained open after 72 hr of competition (bars denote means ± S.E. for $n = 3$). These occurrence frequency data are normalized to the total area of blocked versus open flow in the microfluidic devices, as determined by the presence or absence of fluorescent bead tracks. There is no significant difference in wild type occurrence in regions in which flow is unobstructed and in regions in which flow is blocked (two-sample $t = 0.995$, $df = 4$, p=0.376), but the Δ*pelA* strain is significantly more likely to occur in regions in which flow is blocked at a p<0.05 threshold with Bonferroni correction for two pairwise comparisons (two-sample $t = 3.60$, $df = 4$, p=0.0227). (C) Biofilm growth of wild-type *P. aeruginosa* PA14 (gray) and the Δ*pelA* mutant (white) in monoculture in planar flow chambers under different shear stress exposure treatments (bars denote means ± S.D. for $n = 5$–10).

The following source data and figure supplements are available for figure 3:

**Source data 1.** Occurrence of WT and Pel-deficient *P. aeruginosa* in areas with flow blocked versus areas with flow open.

**Figure supplement 1.** Streamer structures produced by wild-type *P. aeruginosa* PA14 (green) in microfluidic chambers with complex flow profiles do not capture large numbers of co-cultured Δ*pelA* mutants (red) over 72 hr of biofilm growth (black circles are column obstacles).

**Figure supplement 2.** Analysis procedure for correlating local flow and accumulation of Δ*pelA* and wild-type cells.

**Figure supplement 3.** Change in frequency of WT cells from a 1:1 starting population with Δ*pelA* with and without flow.

**Figure supplement 3—source data 1.** Comparison of competition in simple chambers with or without flow.

and only occurs after clogging by the wild type. This result is consistent with prior indications that Pel-producers have higher shear tolerance than Pel-deficient cells (*Colvin et al., 2012*, *2011*), which we confirmed in our system by growing each strain in isolation under varying shear stress in planar chambers (*Figure 3C*). Further supporting our interpretation, the Δ*pelA* strain can outcompete the wild type when the two are grown together in planar microfluidic chambers in the absence of flow (*Figure 3—figure supplement 3*). This experiment approximates the clogged areas of complex flow environments, and the results explain how – on spatial scales encompassing areas of high flow and low or no flow – the wild-type and the Δ*pelA* strains can coexist.

## Conclusions

Biofilm growth is ubiquitous in porous microenvironments and often causes clogging in natural and industrial contexts, including soil beds and water filtration systems (*Knowles et al., 2011*). Here, we have shown that the clogging process can dramatically influence population dynamics within

growing biofilms by generating a feedback between hydrodynamic flow, biofilm spatial architecture, and competition (*Coyte et al., 2016*). Our findings suggest that when *P. aeruginosa* wild type and Δ*pelA* mutants experience irregular flow in heterogeneous environments, wild-type biofilm formation causes partial clogging, regionally reducing local flow speed. The lack of flow generates favorable conditions for the Δ*pelA* strain, whose biofilms would otherwise be removed by shear forces, presumably enabling it to proliferate locally if sufficient nutrients for growth diffuse from other areas of the chamber in which medium continues to flow (*Bottero et al., 2013*).

  *P. aeruginosa* is notorious as an opportunistic pathogen of plants and animals, including humans (*Xavier, 2016*). It also thrives outside of hosts, for example, in porous niches such as soil (*Fierer et al., 2007*). Despite the well-documented ecological benefits of matrix secretion during biofilm formation, environmental and clinical isolates of *P. aeruginosa* exhibit considerable variation in their production of matrix components, including loss or overexpression of Pel (*Mann and Wozniak, 2012*; *Chew et al., 2014*). Our results offer an explanation for natural variation in the ability of *P. aeruginosa* to produce extracellular matrix, particularly among bacteria in porous microhabitats: the evolutionary stable states of extracellular matrix secretion vary with the topographical complexity of the flow environment in which the bacteria reside.

## Materials and methods

### Strains

All strains are derivatives of *Pseudomonas aeruginosa* PA14 (RRID:WB_PA14). Wild-type PA14 strains constitutively producing fluorescent proteins (*Drescher et al., 2013*) were provided by Albert Siryaporn (UC Irvine), and they harbor genes encoding either EGFP or mCherry under the control of the $P_{A1/04/03}$ promoter in single copy on the chromosome (*Choi and Schweizer, 2006*). The Δ*pelA* strain was constructed using the lambda red system modified for *P. aeruginosa* (*Lesic and Rahme, 2008*).

### Liquid growth rate experiments

To determine maximum growth rates and the potential for fluorescent protein production to cause fitness differences, bacterial strains were grown overnight in M9 minimal medium with 0.5% glucose at 37°C. Overnight cultures were back-diluted into minimal M9 medium with 0.5% glucose at room temperature and monitored until their optical densities at 600 nm were ~0.2, corresponding to logarithmic phase. Cultures were back diluted again into minimal M9 medium with 0.5% glucose and transferred to 96-well plates at room temperature. This experiment was repeated for four biological replicates (different overnight inoculation cultures), each repeated for six technical replicates (different wells within a 96-well plate). Measurements of culture optical density at 600 nm were taken once per 10 min until saturation, corresponding to stationary phase. Matlab (Natick, MA) curve fitting software was used to calculate the maximum growth rate of each strain (wild type [GFP]: 0.00716 h$^{-1}$, wild type [mCherry]: 0.00733 h$^{-1}$, Δ*pelA* [mCherry]: 0.00842 h$^{-1}$). These experiments confirmed that the fluorescent protein markers had no measurable effect on growth rates and thus did not contribute to competitive outcomes in our experiments.

### Microfluidics and competition experiments

Microfluidic devices consisting of poly(dimethylsiloxane) (PDMS) bonded to 36 mm x 60 mm glass slides were constructed using standard soft photolithography techniques (*Sia and Whitesides, 2003*). We used planar microfluidic devices with no obstacles to simulate environments with simple parabolic flow profiles, and we used devices with PDMS pillars interspersed throughout the chamber volume to simulate environments with complex (i.e. irregular, non-parabolic) flow profiles. The size and spatial distributions of these column obstacles were determined by taking a cross-section through a simulated volume of packed beads mimicking a simple soil environment. Flow rates through these two chamber types were adjusted to equalize the average initial flow velocities, although the local flow velocity within each chamber varied as biofilms grew during experiments (see main text).

  For all competition experiments, bacterial strains were grown overnight. The following morning, aliquots of the overnight cultures were added to Eppendorf (Hamburg, Germany) tubes, and their

optical densities were equalized prior to preparation of defined mixtures of wild-type and $\Delta pelA$ cells. 100 µL volumes of the wild-type strain alone, the $\Delta pelA$ strain alone, or mixtures of the two strains (for competition experiments), were introduced into microfluidic chambers using 1 mL syringes and Cole-Parmer (Vernon Hills, IL) polytetrafluoroethylene tubing (inner diameter = 0.30 mm; outer diameter = 0.76 mm). After 3 hr, fresh tubing connected to syringes containing fresh minimal M9 medium with 0.5% glucose were inserted into the inlet channels. The syringes (3 mL BD Syringe, 27G; Becton, Dickinson and Co.; Franklin Lakes, NJ) were mounted onto high-precision syringe pumps (Harvard Apparatus; Holliston, MA), which were used to tune flow speeds according to empirical measurements of flow speeds in soil (*Heath, 1983*). In our experiments, the average flow speed was 150–200 µm/s, unless noted otherwise. In *Figure 3C*, to alter shear, we varied the average flow speed; shear was estimated using standard calculations for surface shear stress under fluid flow: $\tau(y) = \alpha \frac{\partial u}{\partial y} = \frac{\Delta p}{L} \frac{H}{2}$ where $\tau$ is the shear stress, $y$ is the height above the surface (evaluated in this case for $y = 0$), $\alpha$ is the dynamic viscosity of the fluid, and $u$ is the fluid flow velocity field, calculated for a rectangular channel in terms of the pressure decrease $\Delta p/L$ across the length $L$ and height $H$ of the channel. The pressure decrease was calculated for our channel dimensions and flow rates using previously published results (*Fuerstman et al., 2007*). Biofilms were grown at room temperature. It should be noted that microfluidic experiments in the obstacle-containing chambers experience a high failure rate, in which no biofilms appear to grow after the 72 hr period of the experiment. No data could be extracted from such chambers, which were omitted from analysis. This problem was overcome by performing the experiment at high replication. Sufficient data were thus collected to populate the relevant panels in *Figures 1* and *3* of the main text. In the case of competition experiments, one replicate was defined as the output from one independently inoculated microfluidic chamber (e.g. *Figure 1A*). For experiments in which biofilm growth was measured as a function of flow-mediated shear stress (*Figure 3C*), one replicate was defined as the output from one imaging location within a microfluidic chamber, with two to three locations per chamber being sampled.

## Design of soil-mimicking microfluidic chambers

To obtain spatial patterns of column obstacles that mimic soil or sand, we first generated a 3D model of packed spheres. The centers of the spheres were positioned such that they had equal radii of 1 (arbitrary units), in a close-packed arrangement. Soil grains, however, are not all the same size. To include heterogeneity in sphere size in our model, we adjusted each sphere's radius using uniformly distributed random numbers to generate a range of sphere radii varying from 0.4 to 1.0. For a plane that is oblique to any of the symmetry planes defined by the centers of the spheres, we generated a cross-section through the 3D packed-sphere model. This cross-section of the spheres was used to define the borders of the columns in our soil-mimicking microfluidic devices. To convert the arbitrarily sized spheres from the 3D model to the actual sizes of physical columns in our microfluidic chambers, we chose column radii that varied from 80 to 200 µm, corresponding to particle sizes of fine- and medium-grain sand.

## Microscopy and image analysis

Mature biofilms were imaged using a Nikon (Tokyo, Japan) Ti-E inverted microscope via a widefield epifluorescence light path (using a 10x objective) or a Borealis-modified Yokogawa CSU-X1 (Tokyo, Japan) spinning disk confocal scanner (using a 60x TIRF objective). A 488-nm laser line was used to excite EGFP, and a 594-nm laser line was used to excite mCherry. Quantification of biofilm composition was performed using Matlab and Nikon NIS Elements analysis software (*Drescher et al., 2014*). Imaging of biofilms could only be performed once for each experiment, precluding time-series analyses, due to phototoxicity effects after multiple rounds of imaging. Phototoxicity was a particularly notable issue here due to dimness of the fluorescent proteins in *P. aeruginosa*, which made long exposures necessary to capture images of sufficient quality for later analysis. For this reason, we opted for inferential population dynamics analysis as described in the main text.

## Effluent measurements

To measure strain frequencies in the biofilm effluent of planar chambers (*Figure 2C*), 1:1 strain mixtures of wild-type and Δ*pelA* cells were prepared and inoculated into simple flow chambers according to the procedure outlined above for competition experiments. At 0, 24, and 48 hr, 5 μL samples were collected from the microfluidic chamber outlet tubing, mixed vigorously by vortex, and plated onto agar in serial dilution. After overnight growth at 37°C, plates were imaged with an Image Quant LAS 4000 (GE Healthcare Bio-Sciences; Pittsburgh, PA). Cy3 and Cy5 fluorescence settings were used for EGFP and mCherry excitation, respectively. Image Quant TL Colony Counting software was used to measure the relative abundance of each strain.

## Flow tracking experiments

1:1 mixtures of the wild-type and the Δ*pelA* mutant were prepared and introduced into obstacle-containing flow chambers according to the procedure described above. Minimal M9 medium with 0.5% glucose was introduced into the chambers for 72 hr as described above. The entire chamber was then imaged using widefield epifluorescence microscopy to document the locations of wild-type and Δ*pelA* cell clusters. Subsequently, the influent syringes were replaced with syringes containing yellow-green fluorescent beads (sulfate-modified, diameter = 2 μm; Invitrogen; Carlsbad, CA) at a concentration of 0.3%, and bead suspensions were flowed into the microfluidic chambers. To determine the presence or absence of flow with respect to the spatial distributions of wild-type and Δ*pelA* cells, and to obtain large images for statistics, the entire chamber was imaged with a 1 s exposure time, over which traveling beads were captured as streaks. It should be noted that this experiment also has a high failure rate due to the sensitivity of the microfluidic chambers to removal and re-insertion of syringes, and required optimization to execute successfully. Custom Matlab code was written to correlate the presence or absence of fluid flow with the accumulation of wild-type and Δ*pelA* cells. In brief, the positions of the columns were first identified and used to divide the chamber into triangular sampling areas using a network structure in which columns served as nodes and straight lines between column centers served as edges. Within each sampling triangle, the area covered by columns was first removed, and subsequently, the averaged Δ*pelA* and wild-type fluorescence intensities in the remaining area were used to determine if a region had wild type and/or Δ*pelA* accumulation. In parallel, each sampling area was scored for the presence of flow in the corresponding bead tracking images (*Figure 3—figure supplement 2*).

## Data display and statistical tests

In all cases where displayed, bars denote the mean values of the measurements taken, and with the exception of *Figure 3B* and *Figure 1—figure supplement 1*, the error bars denote standard deviations. In *Figure 3B* and *Figure 1—figure supplement 1*, error bars denote standard errors. In *Figure 3B*, we report the results of a two-tailed t-test comparing the wild type occurrence frequency in regions of soil-mimicking chambers where flow was obstructed, versus the wild type occurrence frequency in regions where flow was unobstructed. A second t-test was performed to make the same comparison for the Δ*pelA* cells. The p-values from these tests were evaluated against a critical threshold of p<0.05 adjusted by Bonferroni correction for two pairwise comparisons. Two t-tests were also performed on the data in *Figure 1—figure supplement 1* measuring the maximum growth rates of our strains in liquid culture.

## Acknowledgements

We thank members of the BLB laboratory, as well as Thomas Bartlett and Alvaro Banderas for helpful discussions. This work was supported by the Alexander von Humboldt Foundation (CDN), the Human Frontier Science Program Grant CDA00084/2015-C (KD), the Max Planck Society (KD), the Deutsche Forschungsgemeinschaft Grant SFB987 (KD), the Howard Hughes Medical Institute (BLB), NIH Grant 2 R37GM065859 (BLB), National Science Foundation Grant MCB-0948112 (BLB), National Science Foundation Grant MCB-1344191 (BLB), and the Alexander von Humboldt Foundation, the Max Planck Society, and the Federal Ministry of Education and Research (BLB). JY holds a Career Award at the Scientific Interface from the Burroughs Wellcome Fund.

# Additional information

## Funding

| Funder | Grant reference number | Author |
| --- | --- | --- |
| Alexander von Humboldt-Stiftung | | Carey D Nadell<br>Bonnie L Bassler |
| Human Frontier Science Program | CDA00084/2015-C | Knut Drescher |
| Max-Planck-Gesellschaft | | Knut Drescher<br>Bonnie L Bassler |
| Deutsche Forschungsgemeinschaft | SFB987 | Knut Drescher |
| Howard Hughes Medical Institute | | Bonnie L Bassler |
| National Institutes of Health | 2R37GM065859 | Bonnie L Bassler |
| National Science Foundation | MCB-0948112 | Bonnie L Bassler |
| National Science Foundation | MCB-1344191 | Bonnie L Bassler |

The funders had no role in study design, data collection and interpretation, or the decision to submit the work for publication.

## Author contributions

CDN, DR, Conceptualization, Data curation, Formal analysis, Supervision, Funding acquisition, Validation, Investigation, Visualization, Methodology, Writing—original draft, Project administration, Writing—review and editing; JY, KD, Data curation, Software, Formal analysis, Funding acquisition, Validation, Investigation, Methodology, Writing—original draft, Writing—review and editing; BLB, Conceptualization, Resources, Data curation, Supervision, Funding acquisition, Validation, Writing—original draft, Project administration, Writing—review and editing

## Author ORCIDs

Bonnie L Bassler, http://orcid.org/0000-0002-0043-746X

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
