## [Decision Letter]

Thank you for submitting your article "Flow environment and matrix structure interact to determine spatial competition in *Pseudomonas aeruginosa* biofilms" for consideration by *eLife*. Your article has been reviewed by three peer reviewers, one of whom served as Guest Editor, and the evaluation has been overseen by Richard Losick as the Senior Editor. The reviewers have opted to remain anonymous.

The reviewers have discussed the reviews with one another and the Reviewing Editor has drafted this decision to help you prepare a revised submission.

Summary:

This is a very well written manuscript on biofilm formation and strain co-existence in the bacterium *Pseudomonas aeruginosa*. The article studies the advantage of *P. aeruginosa* producing extracellular polymeric substances in different flow environments. By competing a wild-type producer strain with a non-producing mutant, the authors show that while the wild-type out-competes the mutant in an environment where flow is unobstructed, introducing barriers and creating an irregular flow environment can change these predictions to the advantage of the mutant, leading to co-existence. They argue that this is because shear is reduced in the complex environment, creating small pockets where the mutant can grow without being washed away.

The three reviewers all agree that this work is novel and intriguing. While the dynamics of bacterial biofilms within porous environments has long been studied, how genotypes compete at the pore scale has largely been ignored. Previous work has largely focused on Darcy scale phenomena and lack clear mechanistic insights into the processes that shape community structure. The message of the paper is also important, since it shows that selection pressures in simplified environmental conditions in the lab may change in more complex environments. The data also nicely explain natural diversity in biofilm production phenotypes. Finally, the article is clear and well written. However, the reviewers have made a few recommendations that would greatly improve the manuscript.

Essential revisions:

1) Matrix secretion is interesting from a social evolution perspective because matrix secretion is costly, but can potentially benefit other cells in the vicinity including non-matrix producers. Spatial structure has therefore previously been shown to foster strain co-existence in bacterial communities (e.g. Kerr et al. 2002 Nature, Kim et al. 2008 PNAS, Poltak & Cooper 2011 ISME Journal). The dynamics reported here are somewhat different, since previous studies were conducted in the absence of flow. Liquid flow changes the story substantially because fitness now depends on a strain's ability to stay put, and other secretions – which may be important public goods – are washed away. This social component and a clear comparison to previous literature is not well covered in the paper. The authors mention this aspect here and there, but do not clearly introduce the social principles. This should be changed.

The authors should also be careful in stating that secreted matrix cannot be exploited. First, there is evidence that matrix can efficiently be exploited (Van Gestel et al. 2014 ISME Journal). Second, the authors' results do not disprove exploitation, but simply show that it would not occur in this setup.

2) The fitness advantages of the *pelA* mutant versus the WT in the different scenarios are unclear. Why does the fraction of WT cells decrease when they are inoculated at frequencies larger than 0.6 (even if the effect is not huge)? It seems that this could have many causes: measurements in liquid culture show that *pelA* cells do have a slightly larger growth rate than WT cells, but this difference was not statistically significant. Could the difference between the two genotypes be larger in a biofilm that is not exposed to flow (note the huge error bar in Figure 3)? Or perhaps the architecture of the WT biofilm in porous environments is somehow playing a role? It seems that the basic mechanism behind this observation are not discussed in the manuscript.

There is one experiment that may make the story complete: competing the two strains in a regular microfluidic channel with no flow. In this scenario, biofilm can form, but cells will not be washed away. Will the mutant grow as in liquid (Figure 1—figure supplement 1) or are there other advantages to biofilm growth? Will they grow even more relative to the wild-type than in liquid? The authors already show mono-culture data under this condition (Figure 3). This would substantiate their claim in the conclusion that "Pel non-producers do not appear to directly exploit the matrix material produced by wild type cells, but they can take advantage of the low-shear conditions that wild type matrix producers generate in complex flow environments".

We realize how onerous suggesting additional experiments can be and the lead author is now working in a different lab, but it does seem that having some insight on dynamics of competition on flat surfaces and the mechanisms at play in the porous media would make this a much better paper.

3) We are still left with questions about the fundamental mechanisms behind the empirical observations. Though the tradeoffs between surface colonization and dispersal are mentioned in the introduction, it is not clear whether the pel mutants, which are trapped in low flow regions, have relegated themselves to a low nutrient/low dispersal environment. Do the authors have an idea as to whether the cells trapped within these low velocity regions suffer from decreased reproductive fitness? The authors measured cells from the effluent in the flat microfluidic devices, is there a particular reason why similar observations could not be made in the porous channels?

[Editors' note: further revisions were requested prior to acceptance, as described below.]

Thank you for resubmitting your work entitled "Flow environment and matrix structure interact to determine spatial competition in *Pseudomonas aeruginosa* biofilms" for further consideration at *eLife*. Your revised article has been favorably evaluated by Richard Losick (Senior editor) and a Reviewing editor.

The manuscript has been improved but there are some remaining issues that need to be addressed before acceptance, as outlined below:

How do the data in the new Figure (Figure 3—figure supplement 3) compare to Figure 3 and to growth in liquid? Is there a bigger advantage? The best would also be to use similar plots for all growth comparisons (Figure 1—figure supplement 1, Figure 3 and the new figure).

I apologize for not noticing this earlier, but the details of flow rates appear to be missing. What flow rate are you using by default? And how does this change to you achieve the differences in shear in Figure 3? How do you measure and verify differences in shear?

In your response, you state that "Our experiments directly demonstrate that cells in the flow-blocked regions continue to grow". I do not see these data. You have shown – and this is well-presented in the Results section – that there is a higher concentration of *pelA* mutants in areas with reduced flow. You also show in Figure 1 that the frequency of the wildtype is lower in the complex compared to the planar environment. But you have not provided data to show that the population size of *pelA* mutants is increasing over time in the complex environment. If you have these data, please add them as a supplementary figure. Otherwise, unless I have misunderstood something, the conclusions section needs to be modified: what you state "[flow] enabl[es] [DpelA] to proliferate locally using nutrients diffusing from the bulk liquid phase that are replenished by flow in other regions of the environment" has not been shown here. I think changing the sentence to something like "potentially enabling it to proliferate locally assuming that sufficient nutrients are diffusing from the bulk liquid phase…" would be more fitting and sufficient given your current results.

---

## [Author Response]

*Essential revisions:*

*1) Matrix secretion is interesting from a social evolution perspective because matrix secretion is costly, but can potentially benefit other cells in the vicinity including non-matrix producers. Spatial structure has therefore previously been shown to foster strain co-existence in bacterial communities (e.g. Kerr et al. 2002 Nature, Kim et al. 2008 PNAS, Poltak & Cooper 2011 ISME Journal). The dynamics reported here are somewhat different, since previous studies were conducted in the absence of flow. Liquid flow changes the story substantially because fitness now depends on a strain's ability to stay put, and other secretions – which may be important public goods – are washed away. This social component and a clear comparison to previous literature is not well covered in the paper. The authors mention this aspect here and there, but do not clearly introduce the social principles. This should be changed.*

*The authors should also be careful in stating that secreted matrix cannot be exploited. First, there is evidence that matrix can efficiently be exploited (Van Gestel et al. 2014 ISME Journal). Second, the authors' results do not disprove exploitation, but simply show that it would not occur in this setup.*

We thank the referees for pointing out this idea for addition to our manuscript. We agree in full that the social evolution perspective is important to understanding biofilm matrix production. New sentences have been added in the introduction to address the reviewers' concerns. Our goal was to include the social evolution perspective while not making social evolution the central theme of the manuscript, which aims more specifically to show how flow conditions alter spatial competition in biofilms. We are also more careful now to emphasize that biofilm matrix material is not always privatized, which can lead to public goods dilemmas with cheating cells outgrowing cooperative cells, as was observed in the important van Gestel et al. 2014 ISME J paper from the Kovacs group which is now cited.

New text can be found in the Introduction section.

*2) The fitness advantages of the pelA mutant versus the WT in the different scenarios are unclear. Why does the fraction of WT cells decrease when they are inoculated at frequencies larger than 0.6 (even if the effect is not huge)? It seems that this could have many causes: measurements in liquid culture show that pelA cells do have a slightly larger growth rate than WT cells, but this difference was not statistically significant. Could the difference between the two genotypes be larger in a biofilm that is not exposed to flow (note the huge error bar in Figure 3)? Or perhaps the architecture of the WT biofilm in porous environments is somehow playing a role? It seems that the basic mechanism behind this observation are not discussed in the manuscript.*

*There is one experiment that may make the story complete: competing the two strains in a regular microfluidic channel with no flow. In this scenario, biofilm can form, but cells will not be washed away. Will the mutant grow as in liquid (Figure 1—figure supplement 1) or are there other advantages to biofilm growth? Will they grow even more relative to the wild-type than in liquid? The authors already show mono-culture data under this condition (Figure 3). This would substantiate their claim in the conclusion that "Pel non-producers do not appear to directly exploit the matrix material produced by wild type cells, but they can take advantage of the low-shear conditions that wild type matrix producers generate in complex flow environments".*

*We realize how onerous suggesting additional experiments can be and the lead author is now working in a different lab, but it does seem that having some insight on dynamics of competition on flat surfaces and the mechanisms at play in the porous media would make this a much better paper.*

We thank the referees for this suggestion. We have performed the requested experiment, which shows clearly that Pel non-producing cells outcompete wild type cells when they are grown together in planar microfluidic chambers with no flow. This result is consistent with the existing logic of our conclusions and provides the additional needed support for our interpretation that the referees were seeking in their request. New text has also been added to describe these results (subsection “Wild type P. aeruginosa obstructs porous environments, generating low-shear regions that Pel deficient mutants can occupy”). We also added a new supplemental figure (Figure 3—figure supplement 3) with the results of the new experiment.

*3) We are still left with questions about the fundamental mechanisms behind the empirical observations. Though the tradeoffs between surface colonization and dispersal are mentioned in the introduction, it is not clear whether the pel mutants, which are trapped in low flow regions, have relegated themselves to a low nutrient/low dispersal environment. Do the authors have an idea as to whether the cells trapped within these low velocity regions suffer from decreased reproductive fitness? The authors measured cells from the effluent in the flat microfluidic devices, is there a particular reason why similar observations could not be made in the porous channels?*

An important point is that our manuscript is concerned with local competition for space and resources in the biofilm environment: we are exploring how the stable relative abundances of wild type and matrix mutants change as a function of environmental condition. The fact that Pel mutants can locally coexist with wild type cells in the complex flow environment, due to the clogging which reduces flow, whereas they cannot coexist with wild type in the simple flow environment, means by definition that Pel mutants have a higher reproductive fitness under complex flow conditions than they do under simple flow conditions. This result and its underlying causes are the core conclusions of our paper. Regarding nutrient status within flow-blocked areas, we note also that even though bulk transport of nutrients has been reduced or terminated, diffusion of nutrients from neighboring regions continues. Our experiments directly demonstrate that cells in the flow-blocked regions continue to grow.

We have modified existing text and added new sentences in the Introduction section and Conclusion section to emphasize that we are studying within-biofilm competition effects.

[Editors' note: further revisions were requested prior to acceptance, as described below.]

*The manuscript has been improved but there are some remaining issues that need to be addressed before acceptance, as outlined below:*

*How do the data in the new Figure (Figure 3—figure supplement 3) compare to Figure 3 and to growth in liquid? Is there a bigger advantage? The best would also be to use similar plots for all growth comparisons (Figure 1—figure supplement 1, Figure 3 and the new figure).*

We appreciate the request for clarification here. The data shown in Figure 3—figure supplement 3 are changes in wild type population frequency from an initially 1:1 mixed population, i.e., with wild type at an initial frequency of 0.5. These data are most directly comparable to Figure 1, except only one initial population composition is shown in Figure 3—figure supplement 3, rather than nine as in Figure 1. We believe it is important to retain the different plot formats for the three figures mentioned because each plot conveys a different type of information. Figure 1—figure supplement 1 shows growth rate as extracted from time series data of growth in liquid culture. Figure 3 shows the absolute biomass accumulation after a fixed period of biofilm growth under different shear conditions, and lastly, the new Figure 3—figure supplement 3 shows changes in the relative abundance of wild type versus the pelA mutant in biofilms with no flow. To make it clear why the graphs are different, we have now provided additional detail in the figure legend of Figure 3—figure supplement 3 to explain exactly what that graph depicts.

*I apologize for not noticing this earlier, but the details of flow rates appear to be missing. What flow rate are you using by default? And how does this change to you achieve the differences in shear in Figure 3? How do you measure and verify differences in shear?*

In response to the Editor’s question, we have included exact information regarding flow (subsection “Microfluidics and competition experiments”). Shear rates in the planar flow chambers are calculated on the basis of flow rate, rather than being measured directly (which cannot be done in our apparatus). As such they are estimates, but importantly, the estimate of shear under each condition is identical irrespective of which strain is being tested. Thus, the inter-strain comparison (our point of interest) remains correct.

In the revised text, we wrote: “In our experiments, the average flow speed was 150-200 µm/sec, unless noted otherwise. In Figure 3, to alter shear, we varied the average flow speed; shear was estimated using standard calculations for surface shear stress under fluid flow: τ(y)=α∂u∂y= �pL H2 where τ is the shear stress, y is the height above the surface (evaluated in this case for y=0), α is the dynamic viscosity of the fluid, and u is the fluid flow velocity field, calculated for a rectangular channel in terms of the pressure decrease �p/L across the length *L* and height *H* of the channel. The pressure decrease was calculated for our channel dimensions and flow rates using previously published results. Lab Chip *7*, 1479-1489.)”

*In your response, you state that "Our experiments directly demonstrate that cells in the flow-blocked regions continue to grow". I do not see these data. You have shown – and this is well-presented in the Results section – that there is a higher concentration of pelA mutants in areas with reduced flow. You also show in Figure 1 that the frequency of the wildtype is lower in the complex compared to the planar environment. But you have not provided data to show that the population size of pelA mutants is increasing over time in the complex environment. If you have these data, please add them as a supplementary figure. Otherwise, unless I have misunderstood something, the conclusions section needs to be modified: what you state "[flow] enabl[es] [DpelA] to proliferate locally using nutrients diffusing from the bulk liquid phase that are replenished by flow in other regions of the environment" has not been shown here. I think changing the sentence to something like "potentially enabling it to proliferate locally assuming that sufficient nutrients are diffusing from the bulk liquid phase…" would be more fitting and sufficient given your current results.*

In all of our frequency experiments, the total is 100%, thus if one strain decreases in frequency, the other must have increased in frequency. In the case of complex flow chambers, when the wild type strain decreased in frequency relative to the initial population composition, the *pelA* mutant had therefore increased in population size disproportionately relative to the wild type. This interpretation was the basis for our text statement. However, in light of the confusion this experiment/text has caused, we agree with the editor, that altering our concluding sentence to shift the tone toward interpretation, rather than demonstration, is more appropriate. We have made the suggested change in the main text (Conclusions section). We used the word presumably rather than the word potentially (presumably enabling it to proliferate).